# Deep-Learning-Based Classification of Cyclic-Alternating-Pattern Sleep Phases

**DOI:** 10.3390/e25101395

**Published:** 2023-09-28

**Authors:** Yoav Kahana, Aviad Aberdam, Alon Amar, Israel Cohen

**Affiliations:** 1Andrew and Erna Viterbi Faculty of Electrical & Computer Engineering, Technion—Israel Institute of Technology, Technion City, Haifa 3200003, Israel; aamar@ef.technion.ac.il; 2AWS AI Labs, Amazon, Haifa 3760105, Israel; aaberdam@amazon.com

**Keywords:** cyclic alternating pattern (CAP), time-frequency analysis, deep neural networks, convolutional neural network (CNN), CAP sleep database (CAPSLPDB), electroencephalography (EEG), sleep

## Abstract

Determining the cyclic-alternating-pattern (CAP) phases in sleep using electroencephalography (EEG) signals is crucial for assessing sleep quality. However, most current methods for CAP classification primarily rely on classical machine learning techniques, with limited implementation of deep-learning-based tools. Furthermore, these methods often require manual feature extraction. Herein, we propose a fully automatic deep-learning-based algorithm that leverages convolutional neural network architectures to classify the EEG signals via their time-frequency representations. Through our investigation, we explored using time-frequency analysis techniques and found that Wigner-based representations outperform the commonly used short-time Fourier transform for CAP classification. Additionally, our algorithm incorporates contextual information of the EEG signals and employs data augmentation techniques specifically designed to preserve the time-frequency structure. The model is developed using EEG signals of healthy subjects from the publicly available CAP sleep database (CAPSLPDB) on Physionet. An experimental study demonstrates that our algorithm surpasses existing machine-learning-based methods, achieving an accuracy of 77.5% on a balanced test set and 81.8% when evaluated on an unbalanced test set. Notably, the proposed algorithm exhibits efficiency and scalability, making it suitable for on-device implementation to enhance CAP identification procedures.

## 1. Introduction

Detecting sleep stages is essential for understanding and improving sleep quality and identifying and addressing many sleep-related pathologies. Especially the cyclic alternating pattern (CAP) is considered a key concept in evaluating the sleep process [1]. CAP is divided, generally, into two main phases by the distinction between cerebral activation (A-phase) and de-activation (B-phase) modes [2]. Beyond being a physiological phenomenon, CAP is considered a reliable marker of sleep instability [3] as it can correlate with several sleep-related pathologies [4]. Consequently, accurate detection of the CAP phases has a crucial role in a sleep diagnosis. Traditionally, sleep analysis relies on polysomnography (PSG) and is conducted by trained physicians and healthcare professionals in sleep laboratories. This approach poses significant challenges in terms of practicality and clinical applicability. The manual assessment process is labor-intensive, time-consuming, and susceptible to human fatigue, subjectivity, and potential errors. The development and implementation of such automated tools not only enhance the reliability and precision of sleep diagnosis but also have the potential to streamline clinical workflows, reduce healthcare costs, and ultimately improve patient care and outcomes.

To streamline and facilitate this vital process, various methods were proposed over the years to automate the detection of the CAP phases [5,6,7,8,9,10,11,12,13,14,15,16]. Yet, a great majority of the current methods rely on (1) hand-crafted feature extraction, which may not capture all relevant information of the data, and (2) traditional machine-learning-based approaches, rather than taking advantage of the deep learning tools which have been increasingly used in recent years [17] due to their high-performance and proven success in a broad spectrum of tasks [18,19,20]. To bridge this gap, we aim to leverage the capabilities of deep convolutional neural networks (CNN) for classifying CAP phases. Our approach is motivated by the remarkable strides that CNNs have made in recent years [21], achieving state-of-the-art performance in a wide range of tasks and applications, including image classification [22], object tracking [23], text detection [24], speech and natural language processing [25], and others.

This work proposes an end-to-end fully automatic CNN-based method for classifying CAP phases. For this goal, we present an algorithm consisting of three main stages (Figure 1). Firstly, we analyze each second of the long EEG signal as a distinct prolonged 1D-EEG segment, considering the contextual information of the signal. Next, we transform each time segment into its time-frequency representation (TFR). This representation is highly suitable for classifying CAP phases due to the non-stationary nature of the signals, and it allows us to utilize a CNN-based architecture effectively. The TFR is treated as a 2D image and fed into a convolutional neural network that classifies it as either an A-phase or a B-phase sample. Our experimental results demonstrate that we achieve state-of-the-art performance even with the compact ResNet18 architecture [26], which can be efficiently run on-device.

We investigated the usage of several time-frequency transforms and show that the commonly used spectrogram, which relies on the short-time Fourier transform (STFT), is inadequate for the CAP classification task, likely due to its limited time-frequency resolution [27]. Alternatively, we demonstrate that adopting Wigner–Ville-distribution (WVD)-based transformations, which in many cases capture the time-varying frequency content of the signals more accurately [28], significantly enhances the results.

Furthermore, akin to human analysis, which considers the vicinity and context of the EEG signal, our method incorporates extended windows to extract and leverage contextual information from the signal. We have conducted experiments to examine the effect of using various window sizes for improving outcomes and show that involving the sequential information of the EEG signal is crucial to classify the CAP. Finally, to improve the generalization of our model and reduce overfitting [29,30,31], we used data augmentation specially designed to preserve the time frequency and the reasonable structure of the EEG time-frequency representation.

Testing of the proposed method was conducted over the PhysioNet’s public Cap Sleep Database (CAPSLPDB) [2,32], considered a benchmark database for CAP identification and classification research. A thorough experimental study shows our algorithm achieves state-of-the-art performance on the CAP Sleep Database, reaching an accuracy of 77.5% on a balanced test set and 81.8% when evaluated on an unbalanced test set.

To summarize, the main contributions of our work are:A fully automatic classifier of cyclic alternating pattern (CAP) signals, based on a computationally efficient neural network, which therefore can be implemented on-device.Extensive experiments demonstrate state-of-the-art performance on a public CAP benchmark database, classifying its A and B phases using only a single EEG signal.An ablation study was conducted to assess the impact of different time-frequency representations, segment sizes, and types of data augmentation.

The remainder of the paper is organized as follows. In Section 2, we provide essential background information on CAP. Section 3 discusses the related work in the field. Section 4 introduces our proposed end-to-end method, while Section 5 outlines the dataset’s description and the employed performance measures. Section 6 then demonstrates the performance of our proposed method, including an ablation study. Finally, in Section 7, we conclude our findings and offer some directions for future research.

## 2. Background

According to the guidelines set by the American Academy of Sleep Medicine (AASM) [33], sleep is typically classified into five stages that characterize sleep’s macrostructure. These stages include Wakefulness (W), Rapid-Eye-Movement (REM), and Non-Rapid-Eye-Movement, which consists of three interior stages (Non-REM S1–S3). In 2001, the concept of cyclic alternating patterns (CAP) was introduced [2] to characterize the microstructure of sleep. CAP represents a periodic EEG activity that occurs during Non-REM sleep and is characterized by cyclic sequences of cerebral activation (A-phase) followed by periods of deactivation (B-phase) [1]. An A-phase period and the following B-phase period define a CAP cycle; at least two CAP cycles are required to form a CAP sequence. Figure 2 demonstrates CAP in sleep. The figure displays data from six distinct EEG channels (C4A1–P4O2), each spanning a duration of 60 s. The A-phase (red) period and the subsequent B-phase (blue) period define a single CAP cycle, while the consecutive cycles collectively define a CAP sequence.

While B-phase is considered to be the background rhythm of the signal, A-phase can be divided into three interior sub-types [34]:A1 is dominated by slow varying waves (low frequencies, 0.5 Hz–4 Hz) with a high amplitude about the typical background, B-phase.A3 is characterized by increasing in frequency (8 Hz–12 Hz) along with decreasing in the amplitude.A2 is a combination of both A1 and A3.

This work focuses on the binary classification of Non-REM sleep into its A and B CAP phases.

## 3. Related Work

In most studies, the standard approach for CAP classification involves using feature extraction techniques to generate input data for a classifier, which aims to differentiate between the A and B phases. The feature extraction is generally based on the distinctions in energy and frequency content between the A and B phases mentioned earlier. For instance, in [5,6], the EEG signal was divided into distinct frequency bands, and the power spectral density (PSD) was computed for each band separately. Subsequently, PSD-based features were extracted to feed various classifiers—Ref. [5] utilized a linear discriminant analysis (LDA) that assumes the data to be produced based on Gaussian distributions [35], while in [6] different supervised and unsupervised classifiers were evaluated, including decision trees, support vector-machines (SVM), k-means clustering, and others. Similarly, Refs. [7,8,9,10] partitioned the EEG signals into different frequency bands, while in their studies, variance indices were utilized as features. As a classifier, a three-layer neural network was employed in [7], while [8] used SVM and [9] utilized the LDA classifier. In [10], all these classifiers were compared to an adaptive boosting (AdaBoost) classifier, resulting in the superiority of the LDA classifier.

In several works [11,12,13,36,37], time-frequency transforms were utilized to address the pre-mentioned distinctions among the CAP phases. Particularly, Refs. [11,13,36,37] employed the wavelet transform, while [12] used the Wigner–Ville distribution (WVD). Nevertheless, in all these works, the time-frequency transforms were used as a temporary representation for hand-crafted feature extraction, similar to the previous studies.

In recent research, there has been an emerging utilization of deep learning (DL) techniques for classifying CAP phases. Primarily, Ref. [15] achieved high performance (82.4%±7.08% accuracy) by employing a long short-term memory (LSTM) network. Nevertheless, it is worth noting that in this work, the neural network was fed by several hand-crafted features, and its outcomes were post-processed to improve performance by the CAP scoring guidelines outlined in [2].

In [14], a one-dimensional convolutional neural network (1D-CNN) was suggested for both CAP classification and sleep macrostructure scoring task. Similarly, Ref. [38] employed a comparable 1D-CNN architecture to classify CAP phases of healthy and sleep-disordered individuals. The raw EEG signal was standardized in their works before feeding the 1D-CNN. The trained model was tested on both balanced and unbalanced test sets. At the same time, the outcomes indicated moderate performance when tested on an unbalanced dataset (52.99% in [14] and 60.59% in [38]). A 1D-CNN was also utilized in [16] but using a significantly more complex model based on the U-Net framework and a gated-transformer module to extract local features and global contexts.

To conduct training and testing, most of the previously mentioned studies [5,6,10,11,12,13,14,15,16,38] used CAPSLPDB. In general, these studies employed datasets comprised of only normal patients for evaluation purposes. Nevertheless, several studies employed datasets that included both normal and disordered subjects [11,15,16,38].

Inspired by the demonstrated effectiveness of deep learning techniques, our objective is to classify the EEG signal into its respective CAP phases by leveraging its time-frequency representation and employing a 2D convolutional neural network (2D-CNN). The subsequent section provides a comprehensive explanation of our proposed method.

## 4. Proposed Method

Our method consists of three key components, which are driven by three primary considerations:**Context**—Incorporating contextual information of the signal to make the prediction more analogous to the human diagnosis, which inherently involves close vicinity analysis.**CAP prior knowledge**—Utilizing the distinct features of the A-phase events, which are characterized by higher energy levels and high-frequency spectral content compared to the B-phase background.**Deep learning**—Employing a CNN-based architecture as a classifier to leverage the CNN’s high-performance capabilities.

The proposed method consists of three main building blocks (Figure 3), which align with these factors. The first component involves pre-processing, where each analyzed 1 s EEG is treated as a more extended time window. This window contains the central part we want to classify, along with the near vicinity of the signal that is added to provide contextual information. Each 1D-EEG segment is transformed into a 2D time-frequency matrix in the second stage. This representation captures the non-stationary signal’s energy and spectral content, which vary over time. From this point, the obtained 2D time-frequency representations are treated as images, and the proposed method adopts a deep learning framework. In line with that, the received images are stacked into 4D tensors, normalized, and augmented to preserve their time-frequency structure. The processed images are finally fed into a CNN-based architecture for training in a supervised manner. Next, we detail each of these components.

### 4.1. Pre-Processing

To address the extended length of full-night input signals, which typically last between 9 and 10 h, we initiate the process with data segmentation. The annotations of the data pertain to each second of the signal; however, considering the sequential nature of EEG data, the analysis of each signal second involves the inclusion of preceding and subsequent seconds, resulting in prolonged EEG segments that incorporate the contextual information of the signal. In this work, we evaluated different window lengths, ranging from the plain 1 s windows lacking contextual information to 11 s ones at most. Utilizing even longer windows appears to over-emphasize the contextual information and poses storage and computational efficiency challenges.

Since each EEG segment is an extension of its central second, for all window lengths, the label assigned to each segment is determined solely by the label of its central second, regardless of the labels of its other components. For instance, a 5 s segment composed of 2 s of B-phase followed by 3 s of A-phase would be labeled as an A-phase segment due to its central A-phase second, while on the other hand, a similar 5 s segment made up of 3 B-phase seconds at the beginning followed by 2 A-phase seconds, would be designated as a B-phase segment due to its B-phase center. These two scenarios are illustrated in Figure 4.

Ultimately, due to the different sampling rates of the signals at CAPSLPDB, which vary across recordings between 100 Hz to 512 Hz, we downsampled each segment to 32 Hz as a significantly lower sampling rate that preserves the frequency content relevant to the CAP phases. Thus, an identical resolution at the analysis is obtained, and the complexity of calculations is significantly reduced.

### 4.2. Time-Frequency Analysis

Time-frequency analysis is applied to the signals to reveal and exploit both spectral structure and temporal changes of the EEG segments, which is essential for the distinction between A and B phases due to their non-stationary nature. Additionally, the transition of the 1D time segments to 2D time-frequency images allows using a CNN-based classifier. In this study, we explored the use of several time-frequency transformations, including spectrograms (SPECs), Wigner–Ville distributions (WVDs), and smoothed pseudo-Wigner–Ville distributions (SPWVDs). Definitions and additional details for these representations are given in the following paragraphs.

#### 4.2.1. Spectrogram (SPEC)

The spectrogram is widely acknowledged as a prevalent method for analyzing time-varying and non-stationary signals. The spectrogram definition is based on the short-time Fourier transform (STFT), as for a signal, x(t), the STFT is defined as
(1)$$\begin{array}{c} X\left(t,f\right)=\int_{-\infty }^{\infty }x\left(t_{1}\right)h^{*}\left(t_{1}-t\right)e^{-j2\pi ft_{1}}\;dt_{1}, \end{array}$$
where h(t) is a window function centered at time *t*. The window function cuts the signal just close to the time *t*, and the Fourier transform will be an estimate locally around this time instant.

The spectrogram, $S_{x}\left(t,f\right)$, is formulated as the squared magnitude of the STFT:(2)$$S_{x}\left(t,f\right)={\left|X\left(t,f\right)\right|}^{2}\;.$$
The spectrogram is the most widely known and commonly used time-frequency transform [28]. It is well understood, easily interpretable, and has fast implementations, e.g., fast Fourier transform. However, its drawbacks are the limited and fixed resolution in time and frequency which is determined by the length of the window h(t) [27].

#### 4.2.2. Wigner–Ville Distribution (WVD)

The Wigner–Ville distribution of a signal x(t) is given by
(3)$$W_{x}\left(t,f\right)=\int_{-\infty }^{\infty }x\left(t+\frac{\tau }{2}\right)x^{*}\left(t-\frac{\tau }{2}\right)e^{-j2\pi f\tau }d\tau \;.$$
The WVD has the best possible concentration in the time-frequency domain [39], and in particular, can attain a perfect localization for pure frequency-modulated signals [40].

However, the notable drawback of the WVD is known as the cross-terms (CTs) [39]. These artifacts arise when the signal contains a mixture of several signal components, which significantly reduces the readability of the time-frequency representation. The origin of CT lies in the non-linear nature of the WVD transform, which causes the superposition of several components to generate not only the desired auto-term (AT) components but also CT. One of the methods to address this problem is the smoothed pseudo-Wigner–Ville distribution method, as explained immediately.

#### 4.2.3. Smoothed Pseudo Wigner–Ville Distribution (SPWVD)

The smoothed pseudo-Wigner–Ville distribution of a signal x(t) can be formulated as the two-dimensional convolution of the Wigner–Ville distribution, $W_{x}\left(t,f\right)$, with a low-pass-nature kernel, Φ(t,f):(4)$$W_{x}^{sp}\left(t,f\right)=W_{x}\left(t,f\right)**\Phi \left(t,f\right),$$
where ∗∗ represents a 2D convolution.

Equation (Equation 4) can be expressed explicitly:(5)$$\begin{array}{c} W_{x}^{sp}\left(t,f\right)=\int_{-\infty }^{\infty }\int_{-\infty }^{\infty }\int_{-\infty }^{\infty }x\left(u+\frac{\tau }{2}\right)x^{*}\left(u-\frac{\tau }{2}\right)\cdot \Phi \left(\nu ,\tau \right)e^{j2\pi (\nu t-f\tau -\nu u)}dud\tau d\nu \\ =\int_{-\infty }^{\infty }\int_{-\infty }^{\infty }A_{x}\left(\nu ,\tau \right)\Phi \left(\nu ,\tau \right)e^{-j2\pi (f\tau -\nu t)}d\tau d\nu , \end{array}$$
where $A_{x}\left(\nu ,\tau \right)$ is called the ambiguity function (AF) and is defined as
(6)$$A_{x}\left(\nu ,\tau \right)=\int_{-\infty }^{\infty }x\left(t+\frac{\tau }{2}\right)x^{*}\left(t-\frac{\tau }{2}\right)e^{-j2\pi \nu t}dt\;.$$
Equation (Equation 4) formulates the SPWVD as a filtered version of the WVD, whereas the last term in (Equation 5) demonstrates that this filtering is achieved through the multiplication of the AF with the low-pass-nature kernel, Φ(ν,τ). This observation can be rationalized considering that the AF can be viewed as a time-frequency (TF) auto-correlation function of the original signal x(t) [41]. As such, it exhibits most properties of a correlation function, including that its modulus is maximum at the origin [42]. As for the multi-component signal case, the total AF consists of both auto-terms neighboring the origin of the time-frequency plane (ν=0 and τ=0) and cross-terms which are mainly located at a time-frequency distance from the origin. This distance depends directly on the separation in time and frequency of the individual components of the signal. Considering this perspective, it is intelligible that low-pass filtering of the AF means suppressing the cross-terms alongside preserving the desired auto-terms. In that way, the SPWVD is an effective method to deal with the WVD cross-term drawback.

Figure 5 demonstrates the abovementioned forms. It shows an example of a 5 s EEG signal taken from CAPSLPDB (patient ’n1’) and its different time-frequency representations. The increased spectral content of the signal, which exists at the low frequencies (up to 2 Hz) and 8 Hz, is reflected from all the different representations. However, it can be seen that Wigner-based representations have a prominent better resolution in time and frequency compared to the spectrogram. Additionally, it presents the interference in visualization caused by the cross-terms when using WVD, and the mitigation to that problem comes by employing an SPWVD technique.

### 4.3. Deep Learning Architecture

In this phase, the 2D time-frequency images generated in previous stages are stacked into tensors and employed as training data for a convolutional neural network. Table 1 outlines the hyperparameters employed in the training process. The chosen model details and further extensions that were executed, such as normalization and augmentation, will be discussed subsequently. A demonstration of the training progress evolution of our model is provided in Appendix A.

#### 4.3.1. Model

We adopted the widely used and well-established ResNet18 architecture [26], which is renowned for its effectiveness in visual classification tasks [43,44]. To align the ResNet18 model with our specific framework, we modified the first and last layers. These modifications were required to accommodate grayscale images as input and to enable a binary classification at the output. Furthermore, we trained the model from scratch, considering the substantial disparities between the training data used to train the ResNet18 originally, ImageNet [19], which is composed of natural images, and our distinct time-frequency “images”.

#### 4.3.2. Normalization

Traditionally, the input data, $x_{i}$, of neural networks is normalized to be $\tilde{x_{i}}$ with zero-mean and of unit standard deviation [45,46], namely
(7)$$\tilde{x_{i}}=\frac{x_{i}-\bar{x_{i}}}{\sigma_{i}}$$
where $\bar{x_{i}}$ is the mean of $x_{i}$ and $\sigma_{i}$ is its standard deviation.

However, to preserve the difference in energy levels between A-phase and B-phase samples, in this work, we selected to divide each input sample constantly by the *mean* standard deviation of the train set samples, namely
(8)$$\tilde{x_{i}}=\frac{x_{i}-\bar{x_{i}}}{\bar{\sigma }},\bar{\sigma }=\frac{1}{N}\sum_{i\in X_{N}}\sigma_{i}$$
where $X_{N}$ denotes the set of all train samples, and $\bar{\sigma }$ is the mean standard deviation of the this set.

#### 4.3.3. Augmentations

To generalize the learned model [29] and to reduce overfitting to train data [30], we employed a series of augmentations for every batch of data loaded. Initially, we evaluated several traditional augmentations commonly used in computer vision tasks, including color-jitter, rotation, and flips, as outlined in [31]. However, these augmentations led to highly inadequate performance, likely due to their strong correlation with natural images, different from the time-frequency “images” being analyzed [47]. Subsequently, we explored the usage of augmentations designed explicitly for our time-frequency image data. Ultimately, we investigated two main augmentation types:Time-shifts: We employed random time-shifts by applying horizontal random cropping to the training data samples. The cropping was restricted to the horizontal axis, i.e., the time domain, to maintain the spectral information of the signals and preserve the distinction between the different phases of CAP, which differ significantly in their spectral characteristics.Time-frequency augmentations (TF-Aug.): A specialized selection of augmentations was utilized to characterize the time-frequency representations effectively. These augmentations were repeatedly applied to each data sample before inputting the neural network. The selected augmentations are:**Noise**: Additive white Gaussian noise (AWGN) with a uniformly distributed standard deviation. Adding noise was specified in [48] as an appropriate and effective augmentation for EEG signals**Gaussian blur:** The time-frequency images were blurred using a Gaussian kernel. This augmentation was randomly applied to the input samples with a probability of p=0.5, meaning that approximately half of the images underwent blurring.**SpecAugment** [49]: A commonly used method for augmenting spectrograms and other time-frequency representations, typically for speech recognition tasks. The augmentation is primarily based on applying random masks to certain frequency bands and time steps in the spectrogram. In this study, we randomly blocked bands up to 5% of image width for time and 3% of image height for frequency.**Crop and Resize:** To imitate extended temporal CAP events, we randomly cropped the images vertically and then resized them back to their original size, slightly stretching the temporal duration of CAP events.

## 5. Materials and Methods

### 5.1. Database Description

The proposed method was developed and evaluated over the publicly available CAP sleep database (CAPSLPDB) [2,32], considered a benchmark for CAP research. The database contains a collection of polysomnographic recordings registered at the Sleep Disorders Center of the Ospedale Maggiore of Parma, Italy. It includes data from a diverse group of 108 patients: healthy individuals and those with various pathological conditions, such as bruxism, insomnia, and others. Each record includes three or more EEG signals and a series of other indicators, such as electrooculogram (EOG), chin and tibial electromyogram (EMG), and ECG signals. Additionally, the database includes accurate CAP annotations corresponding to each second of the signals. The left side of Table 2 summarizes the sample composition per subject in the database. The database exhibits a highly imbalanced distribution, with a significantly higher occurrence of B-phase samples than A-phase events.

### 5.2. Performance Measures

To assess the classification performance under various configurations, we calculated several metrics. These included the number of correctly identified A-phase events (true positives, $t_{p}$), the number of correctly recognized B-phase samples (true negatives, $t_{n}$), as well as the count of samples incorrectly classified as A-phase (false positive, $f_{p}$) or as B-phase (false negative, $f_{n}$). Based on these metrics, we computed accuracy (ACC), precision (PRE), recall (REC), specificity (SPE) and F${}_{1}$ score (F1) using the following expressions:(9)$$ACC=\frac{t_{p}+t_{n}}{t_{p}+t_{n}+f_{p}+f_{n}}\;,PRE=\frac{t_{p}}{t_{p}+f_{p}}\;.$$
(10)$$REC=\frac{t_{p}}{t_{p}+f_{n}}\;,SPE=\frac{t_{n}}{t_{n}+f_{p}}\;.$$
(11)$$F1=\frac{2\cdot t_{p}}{2\cdot t_{p}+f_{p}+f_{n}}\;.$$

Regarding the data splitting for evaluation, in contrast to prior CAP studies [10,11,12,15,50] that employed K-fold cross-validation, we adopted the standard practice of dividing the data into three disjoint subsets: training, validation, and test sets, as seen in various prominent works [19,51,52,53]. The distribution was approximately 80% for training and 10% each for validation and test sets.

### 5.3. Dataset Creation

For this study, we built our dataset using the recordings of the sixteen normal (no pathology) patients, where a single EEG channel was utilized per participant (either the C4-A1 or the C3-A2 channel). Construction of the dataset from the long full-night EEG signals was performed through several steps. To ensure the spread of the samples in the training, validation, and test sets throughout the entire recording, each full-night EEG signal was divided into non-overlapping 300 s segments. The first 240 s of each segment were assigned to the training set, the subsequent 30 s were allocated to the validation set, and the remaining 30 s were designated as the test set. Subsequently, since our proposed algorithm takes each second as a prolonged time window comprising contextual information, removing the seconds at the edges of the resulting segments is essential to achieve a complete separation between the training, validation, and test sets. Ultimately, an appropriate percentage of B-phase samples were randomly removed per recording to achieve a balanced dataset; i.e., the number of A-phase and B-phase samples is equal (see right side of Table 2).

## 6. Numerical Results

We evaluate the performance of the proposed algorithm on the dataset described above and compare it to the results of prior studies. An ablation study was also carried out through a series of experiments to investigate the following aspects:The influence of utilizing various time-frequency representations as input to the classifiers.The impact of incorporating the EEG signal context information by using segments with an increased duration.The determination of appropriate data augmentations strategies for analyzing EEG signals within the proposed framework.

In the first experiment, we compared the three time-frequency representations: spectrogram (SPEC), WVD, and SPWVD. In this experiment, only random time-shifts were applied without further augmentations. For the spectrogram, we used the Hanning window with support of 20% of single data sample length and maximal overlap between subsequent windows; namely, the window moves only one sample each time. As for the WVD and SPWVD, we used the built-in MATLAB function with its default parameters. Figure 6 presents the accuracy of the different representations using increasing window size from 1 s to 11 s (incremented by 2 s).

**Time-Frequency Representation Influence.** The results in Figure 6 and Table 3 clearly show that utilization of Wigner-based transforms (WVD and SPWVD) is much better compared to a Fourier-based spectrogram (SPEC). This is evident in the higher accuracy obtained by WVD and SPWVD for all window sizes greater than 1 s, with only SPEC achieving better accuracy compared to WVD for the 1 s window size (65.65% compared to 59.07% for WVD). Nevertheless, SPWVD still outperforms SPEC at the 1 s case, with an accuracy of 66.75%. As mentioned above, the prominence of Wigner-based transforms over the spectrogram can likely be attributed to the limitations of STFT in terms of time-frequency resolution. In contrast, WVD provides an optimal concentration in the time-frequency domain.

In general, the differences between WVD and SPWVD are negligible. This similarity reveals that cross-terms, which substantially hinder human interpretability, are considered tolerable by the trained model, which learns to deal with these components during the training process and may even leverage them as supplementary features. Yet, the 1 s window case is an exception to this similarity, where WVD demonstrates a notably lower accuracy.

**Contextual information.** Additional valuable insight depicted from Figure 6 is that increasing the window size and the additional contextual information it provides significantly improves the accuracy across all time-frequency representations. The improvement in accuracy reaches approximately 10% for STFT and SPWVD to roughly 20% for WVD. Nevertheless, this trend seems to plateau at the increment from 9 s to 11 s window sizes for WVD and SPWVD, possibly due to an excessive proportion of contextual information about the central primary data. Further increasing the window size beyond 11 s was not explored in this study due to the required prolonged training time.

**Data Augmentations.** In this experiment, three data augmentation techniques were compared to determine an appropriate augmentation strategy for the proposed framework. Additionally, the no-augmentation case was tested as a benchmark. In all cases, the SPWVD was utilized as the time-frequency representation. The evaluated augmentation types were as follows:TF-augmentation: TF-augmentations are applied solely. As described above, these augmentations are designed to maintain the time-frequency structure.Random time-shifts: In this case, the original dataset is augmented by incorporating random time shifts into its samples.TF-augmentations and random time-shifts: Both TF-augmentations and random time-shifts are applied to the dataset.

The results presented in Figure 7 and Table 4 highlight the positive effect of augmenting the primary dataset. The improvement in the accuracy of the trained model, with relation to the non-augmentation case, is observed for window sizes larger than 3 s and ranging from 1 to 2%.

When considering the comparison between the different augmentation techniques, the disparities in accuracy results are inconsequential. However, from the perspective of overall system considerations, it is advantageous to utilize TF-augmentations instead of random time-shifts, since the former does not necessitate the production of extended signals, resulting in improved storage efficiency and reduced computational complexity. It is also noted that using both TF-augmentations and time-shifts does not result in further performance improvements and is, therefore, unnecessary.

### A-Phase Detection

The confusion matrix of the resulting model (for a 9 s segment, SPWVD transform, and TF-augmentations) is shown in Figure 8, which reveals that the true-positive rate (TPR) is similar for the A-phase and B-phase classes, with TPR values of 76.7% and 80.8%, respectively. Table 5 presents performance per A-phase subtype. As the classification is binary, the TPR per subtype refers to the instances that were classified as A-phase generally. The results show that the TPR of A2 and A1 subtypes is significantly higher than the TPR of A3 (85% and 80.6% for A2 and A1, versus 66.2% for A3).

Considering the characteristics of the different A-phase subtypes detailed above, this finding may suggest that the learned model identifies A-phase events primarily as intense in power events (A1) rather than an elevation in the signal’s spectral content (A3). In line with this, A2 events are best identified by the model since they exhibit an increase in both power and frequency.

In summary, Table 6 presents a comparative analysis of our method’s results alongside those obtained by contemporary methods in the field. This table concludes the findings of the numerical results section.

## 7. Conclusions

In this study, we proposed a novel algorithm that employs a convolutional neural network to automatically identify CAP phases through the classification of time-frequency representations. Our approach leverages the sequential structure of the EEG signal, incorporating contextual information into the classification process. Additionally, we developed specially designed data augmentation techniques to preserve the time-frequency structure. Through an ablation study, we assessed the contributions of critical components of our method, including different time-frequency methods, various window sizes, and data augmentation techniques. Extensive experiments on a benchmark database demonstrated the effectiveness of our method, achieving a high-performance accuracy of 77.5% on a balanced test set and 81.8% when evaluated on an unbalanced test set.

Overall, we have developed an end-to-end method employing an efficient CNN, which can be readily implemented on-device, promising significant improvements in clinical procedures. While our model has demonstrated strong performance compared to current methods, there is room for further refinement. Future work should consider training the model on a larger dataset encompassing both healthy and disordered patients. This expansion in data size and diversity is anticipated to improve the model’s generalization and accuracy significantly. Future work may include the exploration of additional backbone networks along with advanced architectures tailored for analyzing serial data and the integration of multi-channel data into the process, ultimately contributing to the advancement of CAP phase identification in clinical practice and sleep medicine.

## Figures and Tables

**Figure 1 entropy-25-01395-f001:**
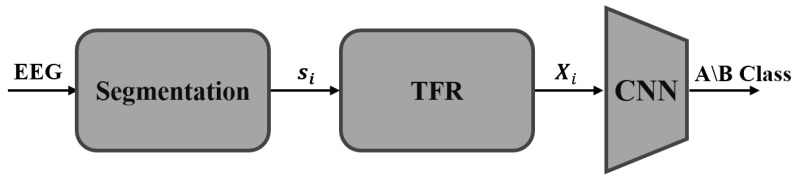
Proposed CAP Classification Workflow: The EEG signal is segmented, transformed into a 2D time-frequency representation (TFR), and fed into a convolutional neural network (CNN) architecture for A/B-phase classification.

**Figure 2 entropy-25-01395-f002:**
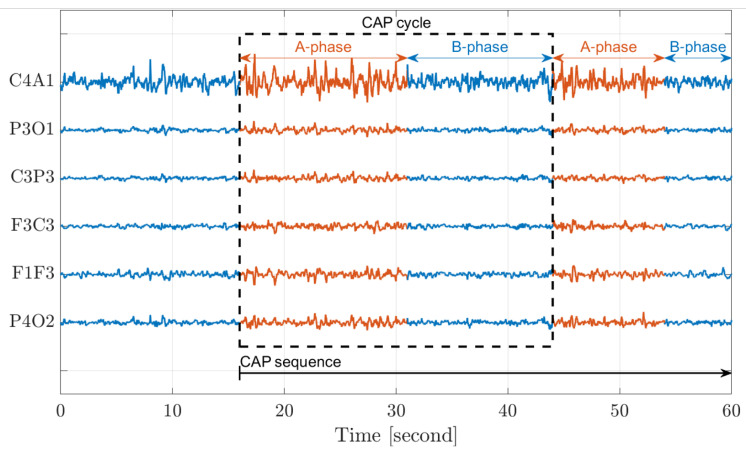
A demonstration of the cyclic alternating pattern (CAP) in sleep.

**Figure 3 entropy-25-01395-f003:**
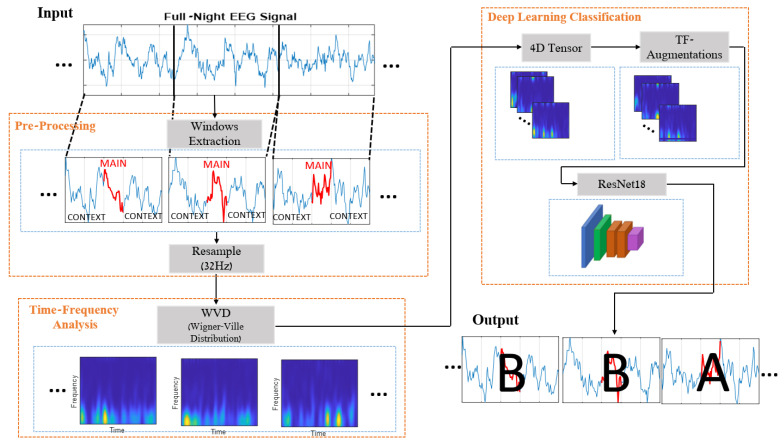
Proposed method scheme.

**Figure 4 entropy-25-01395-f004:**
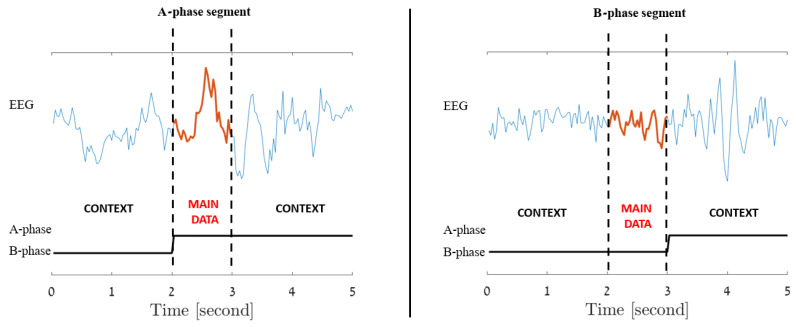
Incorporated Contextual Information: Each signal second extends to include preceding and subsequent seconds, labeled by its central (main data) second. The figure illustrates A-phase (**left**) and B-phase (**right**) 5 s data segments.

**Figure 5 entropy-25-01395-f005:**
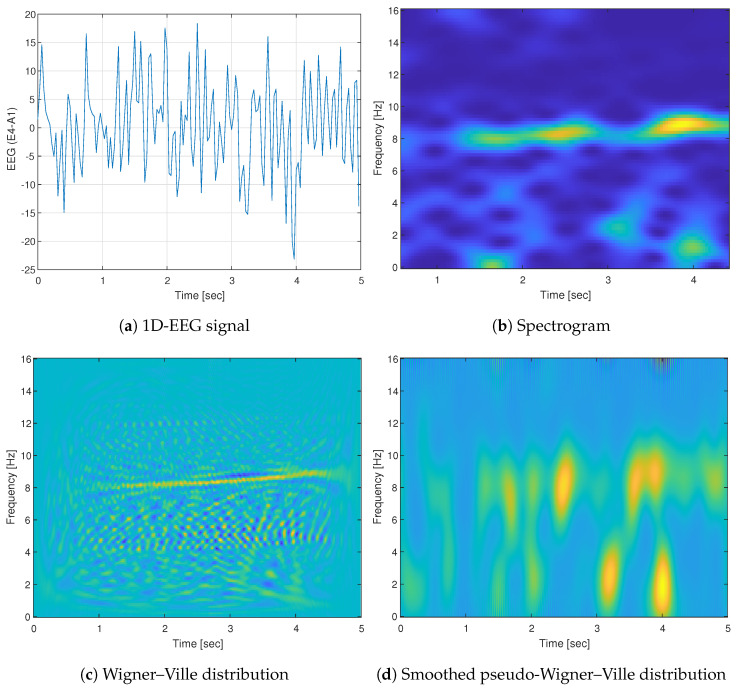
Example of (**a**) 5 s 1D-EEG segment from channel E4-A1 and its corresponding time-frequency representations (TFRs): (**b**) spectrogram (SPEC), (**c**) Wigner–Ville distribution (WVD), and (**d**) smoothed pseudo-Wigner–Ville distribution (SPWVD). The WVD exhibits a distinct energy concentration when compared to SPEC, albeit with the tradeoff of noticeable cross-term patterns.

**Figure 6 entropy-25-01395-f006:**
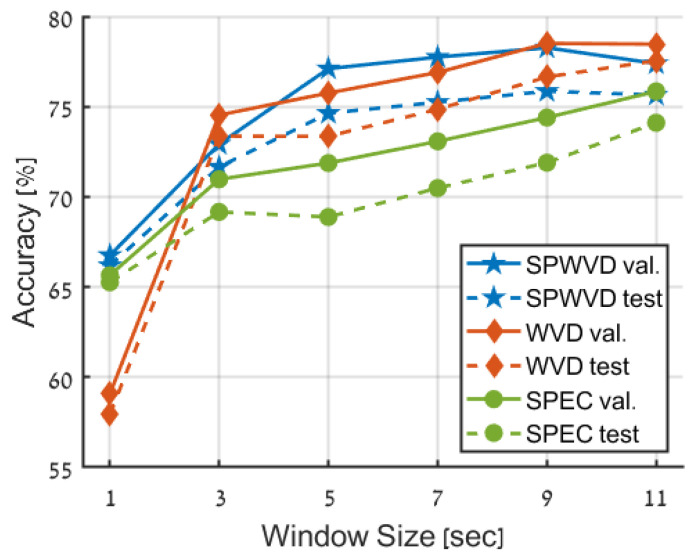
Comparison of performance achieved using different time-frequency representations (TFRs) and window sizes. The blue line corresponds to the SPWVD transform, the red line to the WVD, and the green line to the spectrogram (SPEC). The validation and test data are depicted as solid and dashed lines, respectively.

**Figure 7 entropy-25-01395-f007:**
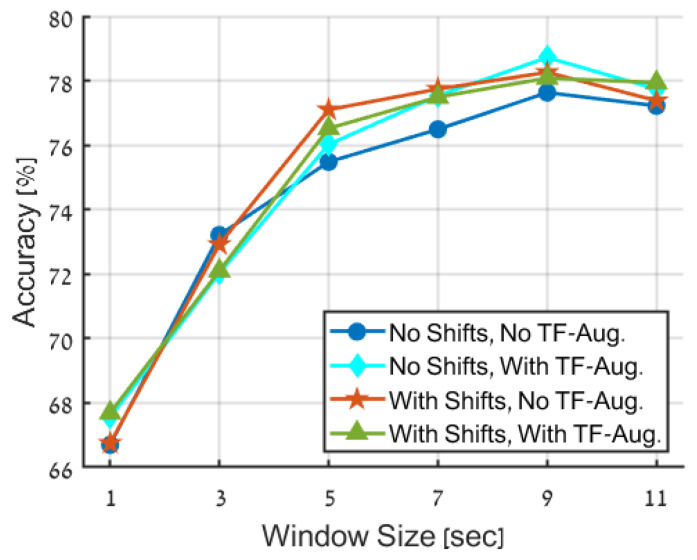
Accuracy comparison of various data augmentation techniques. The figure shows the performance of four strategies: no data augmentation (blue), proposed TF-augmentations only (cyan), random time-shifts only (red), and employment of both time-shifts and TF-augmentation (green).

**Figure 8 entropy-25-01395-f008:**
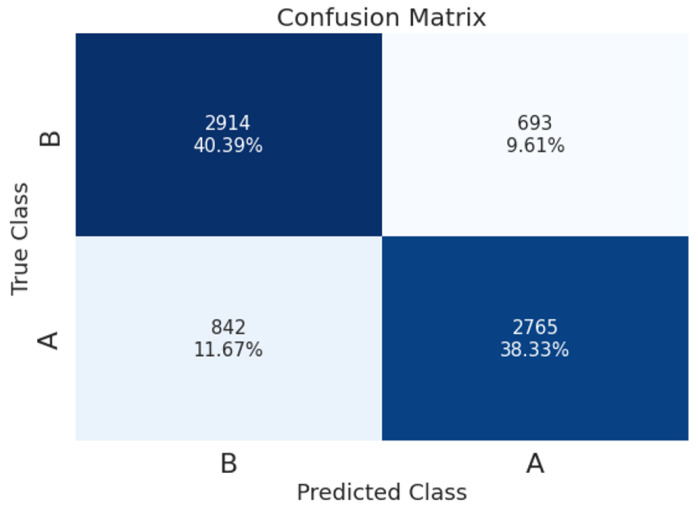
Confusion matrix of CAP detection using 9 s segment’s length, SPWVD, and TF-augmentations.

**Table 1 entropy-25-01395-t001:** Hyperparameters used in the proposed framework.

Hyperparameter	Value
Batch size	256
Loss functions	cross-entropy
Optimizer	SGD
Learning rate	0.001
Momentum	0.9
Epochs	40
Dropout	No

**Table 2 entropy-25-01395-t002:** Total number of samples per healthy subject in the original CAP sleep database (CAPSLPDB) and the corresponding number of samples selected for this study’s dataset. The original CAPSLPDB shows a significantly higher prevalence of B-phase samples than A-phase samples. In contrast, the dataset utilized in this study exhibits a balanced distribution of both A-phase and B-phase classes.

Subject Name	CAPSLPDB (Unbalanced)					Our Dataset (Balanced)				
	$A_{1}$	$A_{2}$	$A_{3}$	Total *A*	B	$A_{1}$	$A_{2}$	$A_{3}$	Total *A*	B
n1	2217	747	1122	**4086**	**21,804**	2063	703	1046	**3812**	**3812**
n2	1115	590	783	**2488**	**12,122**	1036	552	693	**2281**	**2281**
n3	611	597	891	**2099**	**15,451**	550	556	830	**2281**	**2281**
n4	986	356	848	**2190**	**15,030**	928	323	797	**2048**	**2048**
n5	2854	328	620	**3802**	**18,158**	2673	314	586	**3573**	**3573**
n6	1871	970	1401	**4242**	**17,268**	1723	905	1280	**3908**	**3908**
n7	1616	564	479	**2659**	**17,501**	1508	525	438	**2471**	**2471**
n8	949	465	1868	**3282**	**17,028**	914	421	1752	**3087**	**3087**
n9	1036	377	676	**2089**	**18,341**	959	363	641	**1963**	**1963**
n10	1484	326	829	**2639**	**13,351**	1385	282	785	**2452**	**2452**
n11	1724	583	796	**3103**	**15,377**	1640	539	734	**2913**	**2913**
n12	1064	153	573	**1790**	**18,040**	986	139	515	**1640**	**1640**
n13	1628	1037	1017	**3682**	**14,078**	1532	985	955	**3472**	**3472**
n14	1035	1234	1209	**3478**	**15,902**	950	1118	1126	**3194**	**3194**
n15	1449	1046	1244	**3739**	**18,461**	1345	967	1159	**3471**	**3471**
n16	2247	1125	837	**4209**	**17,841**	2110	1041	786	**3937**	**3937**

**Table 3 entropy-25-01395-t003:** Accuracy results (%) for different TFRs and segmentation lengths. In each cell, the upper result refers to the validation set performance, while the lower result refers to the test set performance. The highest results are highlighted within each column, demonstrating the superiority of Wigner-based representations over spectrogram. Additionally, the impact of increased window size is observable.

Method	Window Size					
	1 s	3 s	5 s	7 s	9 s	11 s
SPEC	65.65	70.97	71.87	73.07	74.39	75.84
	65.26	69.16	68.88	70.48	71.89	74.10
WVD	59.07	**74.53**	75.75	76.89	**78.50**	**78.46**
	57.93	**73.36**	73.35	74.85	**76.64**	**77.54**
SPWVD	**66.75**	72.92	**77.10**	**77.74**	78.26	77.38
	**66.19**	71.65	**74.63**	**75.24**	75.85	75.64

**Table 4 entropy-25-01395-t004:** Accuracy results (%) for different data augmentations and extensions of the dataset. In each cell, the upper result refers to the validation set performance, while the lower result refers to the test set performance. In each column, the highest results are highlighted. The results demonstrate the effectiveness of integrating data augmentation. Notably, the highest accuracy (78.72%) was achieved using the proposed TF-augmentations with a 9 s window size.

Dataset’s Composition	Window Size					
	1 s	3 s	5 s	7 s	9 s	11 s
Basic dataset	66.70	**73.21**	75.48	76.49	77.63	77.22
	65.57	71.19	72.90	73.57	75.77	74.06
TF-augmentations	67.55	72.03	76.02	77.53	**78.72**	77.74
	**67.39**	70.08	73.82	74.73	75.76	74.24
Time-shifts	66.75	72.92	**77.10**	**77.74**	78.26	77.38
	66.19	**71.65**	**74.63**	**75.24**	**75.85**	**75.64**
TF-augmentations & time-shifts	**67.69**	72.10	76.52	77.49	78.08	**77.94**
	67.36	71.07	74.20	75.08	75.11	74.87

**Table 5 entropy-25-01395-t005:** True-positive rate (%) per A-phase sub-types and B phase.

Predicted	True			
	A1	A2	A3	B
A	1422	561	782	2914
B	343	99	400	693
**TPR [%]**	**80.6**	**85**	**66.2**	**80.8**

**Table 6 entropy-25-01395-t006:** Summary and comparison between recent studies evaluated on a balanced CAPSLPDB-based dataset. Our method’s results indicated in the table were obtained using the Wigner–Ville distribution (WVD), an 11 s window size, and the proposed time-shift augmentations.

Author	Method	Segment Length [s]	Number of Subjects	Performance Parameter [%] on Validation Set	Performance Parameter [%] on Test Set	Accuracy [%] Evaluated on Unbalanced Test Set
Dhok et al. [12]	Wigner–Ville distribution (WVD), Renyi entropy (RE), support vector machine (SVM)	2	6 patients	ACC = 72.3 PRE = 64.1 REC = 76.8 SPE = 69.2 F1 = 69.9	-	-
Sharma et al. [11]	Wavelet-based features, SVM	2	16 patients	ACC = 75.7 PRE = 75.0 REC = 77.7 F1 = 76.0	-	-
Sharma et al. [13]	Biorthogonal wavelet filter bank (BOWFB), ensemble bagged tree	2	6 patients	ACC = 74.4 REC = 67.53 SPE = 81.3	-	-
Hartmann et al. [15]	Hand-crafted features, long short-term memory (LSTM)	1–3	16 patients	ACC = 82.4±7.1 REC = 75.3±12 SPE = 83.9±8.9 F1 = 57.4±9.6	-	-
Loh et al. [14]	1D-CNN	2	6 patients	ACC = 74.4	ACC = 73.6 PRE = 71.0 REC = 80.3 SPE = 67.0 F1 = 75.3	53.0
Murarka et al. [38]	1D-CNN	2	6 patients	ACC = 76.7	ACC = 78.8 PRE = 82.5 REC = 73.4 SPE = 84.3 F1 = 77.7	60.6
**Our method**	**Spectrogram, Wigner-based representations, ResNet18**	**1–11**	**16 patients**	**ACC = 78.5** **PRE = 78.9** **REC = 77.8** **SPE = 79.3** **F1 = 78.4**	**ACC = 77.5** **PRE = 78.4** **REC = 75.9** **SPE = 79.1** **F1 = 77.1**	**81.8**

## Data Availability

The data presented in this study are available on request from the corresponding author.

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
