# Peer review of "Deep-Learning-Based Classification of Cyclic-Alternating-Pattern Sleep Phases"

_entropy, 2023, doi:10.3390/e25101395_

Round 1

Reviewer 1 Report

Authors propose a deep learning approach to predict different phases of sleep. Here are my comments:

·         In abstract, authors should also mention brief dataset description. Also, more keywords can be added.

·         In introduction, problem definition can be extended. Also, clinical applicability should also be addressed.

·         Figure captions are too long, please consider revising (for example figure 1.)

·         In introduction, 2nd and 3rd paragraphs are the same.

·         Reference text related to Figure 2 is missing.

·         A paragraph that gives an outline of the manuscript structure should be given.

·         In related work section, authors should also specify which studies used the Physionet dataset.

·         Figures should come after they are referenced in the text.

·         In 5.1, for labelling why did the authors choose central second? Details on the decision is not clear in the text.

·         Augmentation methods need more explanation and references.

·         Sentences at line 377 and 378 is unnecessary and can be removed.

·         More evaluation metrics can be considered. This will increase the validity of proposed approach.

·         Conclusion is too short. Authors need to discuss positive aspects of their proposed approach and also limitations.

Reviewer 2 Report

The proposed method has been duplicated, appearing in both Section 4 and Section 5. Section 4 should be omitted.

It is advisable to include figure showing the proposed deep learning model training and validation accuracy vs epochs and the loss vs epochs.

Need to include details about number of epochs and dropout percentages in Table 1.

It is important to expand the range of performance evaluation metrics to encompass metrics such as Precision, Recall, and F1-Score, among others.

Given that the accuracy achieved by the proposed ResNet18 deep learning model is only 77.5% and 81.8%, it is strongly recommended to compare its performance against other well-established deep learning models, such as VGG19 and Xception, to provide a more comprehensive analysis.

The paper is generally well and the correctness of the language is overall good.

Round 2

Reviewer 1 Report

The authors answered my concerns and revised accordingly.